# PRUNING BY ACTIVE ATTENTION MANIPULATION

## ABSTRACT

Filter pruning of a CNN is typically achieved by applying discrete masks on the CNN's filter weights or activation maps, post-training. Here, we present a new filter-importance-scoring concept named pruning by active attention manipulation (PAAM), that sparsifies the CNN's set of filters through a particular attention mechanism, during-training. PAAM learns analog filter scores from the filter weights by optimizing a cost function regularized by an additive term in the scores. As the filters are not independent, we use attention to dynamically learn their correlations. Moreover, by training the pruning scores of all layers simultaneously, PAAM can account for layer inter-dependencies, which is essential to finding a performant sparse sub-network. PAAM can also train and generate a pruned network from scratch in a straightforward, one-stage training process without requiring a pre-trained network. Finally, PAAM does not need layer-specific hyperparameters and pre-defined layer budgets, since it can implicitly determine the appropriate number of filters in each layer. Our experimental results on different network architectures suggest that PAAM outperforms state-of-the-art structured-pruning methods (SOTA). On CIFAR-10 dataset, without requiring a pre-trained baseline network, we obtain 1.02% and 1.19% accuracy gain and 52.3% and 54% parameters reduction, on ResNet56 and ResNet110, respectively. Similarly, on the ImageNet dataset, PAAM achieves $1.06\%$ accuracy gain while pruning $51.1\%$ of the parameters on ResNet50. For Cifar-10, this is better than the SOTA with a margin of 9.5% and 6.6%, respectively, and on ImageNet with a margin of 11%.

## 1 INTRODUCTION

Convolutional Neural Networks (CNNs) LeCun et al. (1989) are used nowadays in a wide variety of computer-vision tasks. Large CNNs in particular, achieve considerable performance levels, but with significant computation, memory, and energy footprints, respectively Sui et al. (2021). As a consequence, they cannot be effectively employed in resource-limited environments such as mobile or embedded devices. It is therefore essential to create smaller models, that can perform well without significantly sacrificing their

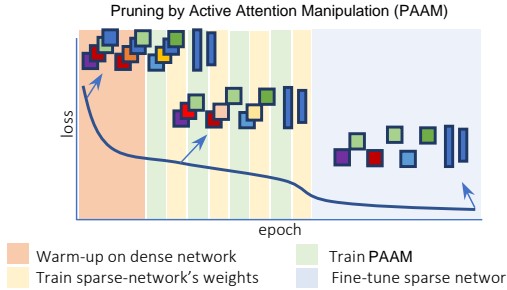

Figure 1: Sensitivity-based filter pruning schedule.

accuracy and performance. This goal can be accomplished by either designing smaller, but performant, network architectures Lechner et al. (2020); Tan & Le (2019) or by first training an over-parameterized network, and sparsifying it thereafter, by pruning its redundant parameters Han et al. (2016); Liebenwein et al. (2020; 2021). Neural-network pruning is defined as systematically removing parameters from an existing neural network Hoefler et al. (2021). It is a popular technique to reduce growing energy and performance costs and to support deployment in resource-constrained environments such as smart devices. Various pruning approaches have been developed, and this has gained considerable attention over the past few years Zhu & Gupta (2017); Sui et al. (2021); Liebenwein et al. (2021); Peste et al. (2021); Frantar et al. (2021); Deng et al. (2020).

Pruning methods are categorized into either unstructured or structured. The first, remove individual weight-parameters, only Han et al. (2016). The second, remove entire groups, by pruning neurons, filters, or channels, respectively Anwar et al. (2017); Li et al. (2019); He et al. (2018b); Liebenwein

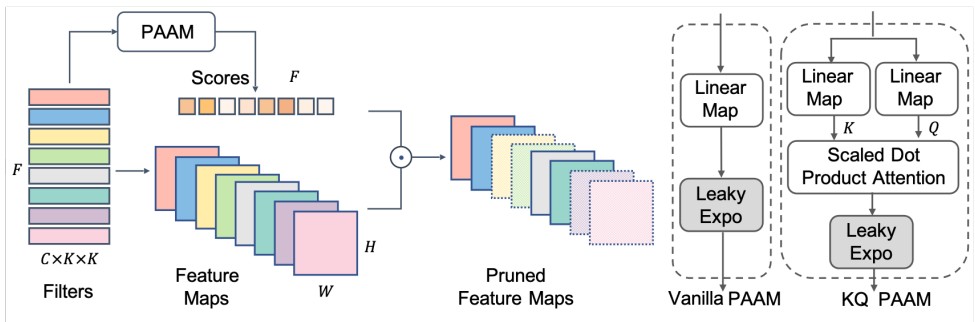

Figure 2: PAAM learns the importance scores of the filters from the filter weights.

et al. (2020). As modern hardware is tuned towards dense computations, structured pruning offers a more favorable balance between accuracy and performance Hoefler et al. (2021). A very prominent family of structured-pruning methods is filter pruning. Choosing which filters to remove according to a carefully chosen importance metric (or filter score) is an essential part of any method in this family.

Data-free methods solely rely on the value of weights, or the network structure, in order to determine the importance of filters. Magnitude pruning, for example, is one of the simplest and most common of such methods. It prunes the filters that have the smallest weight-values in the $l1$ norm. Data-informed methods focus on the feature maps generated from the training data (or a subset of samples) rather than the filters alone. These methods vary from sensitivity-based approaches (which consider the statistical sensitivity of the output feature maps with regard to the input data Malik & Naumann (2020); Liebenwein et al. (2020)), to correlation-based methods (with an inter-channel perspective, to keep the least similar (or least correlated) feature maps Sun et al. (2015); Sui et al. (2021)).

Both data-free and data-informed methods generally determine the importance of a filter in a given layer, locally. However, filter-importance is a global property, as it changes relative to the selection of the filters in the previous and next layers. Moreover, determining the optimal filter-budget for each layer (a vital element of any pruning method), is also a challenge, that all local, importance-metric methods face. The most trivial way to overcome these challenges, is to evaluate the network loss with and without each combination of $k$ candidate filters out of $N$. However, this approach would require the evaluation of $\binom{N}{k}$ subnetworks, which is in practice impossible to achieve.

Training-aware pruning methods aim to learn binary masks for turning on and off each filter. A regularization metric often accompanies them, with a penalty guiding the masks to the desired budget. Mask learning, simultaneously for all filters, is an effective method for identifying a globally optimal subset of the network. However, due to the discrete nature of the filters and binary masks, the optimization problem is generally non-convex and NP-hard. A simple trick of many recent works Gao et al. (2020; 2021); Li et al. (2022) is to use straight-through estimators Bengio et al. (2013) to calculate the derivatives, by considering binary functions as identities. While ingenious, this precludes learning the relative importance of filters among each other. Even more importantly, the on-off bits within the masks are assumed to be independent, which is a gross oversimplification.

This paper solves the above problems by introducing PAAM, a novel end-to-end pruning method, by active attention manipulation. PAAM also employs an $l1$ regularization technique, encouraging filter-score decrease. However, PAAM scores are analog, and multiply the activation maps during score training. Moreover, a proper score spread is ensured through a specialized activation function. This allows PAAM to learn the relative importance of filters globally, through gradient descent. Moreover, the scores are not considered independent, and their hidden correlations are learned in a scalable fashion, by employing an attention mechanism specifically tuned for filter scores. Given a global pruning budget, PAAM finds the optimal pruning threshold from the cumulative histogram of filter scores, and keeps only the filters within the budget. This relieves PAAM from having to determine per layer allocation budgets in advance. PAAM then retrains the network without considering the scores. This process is repeated until convergence. PAAM pipeline is shown in Figures 1-2. Our experimental results show that PAAM yields higher pruning ratios while preserving higher accuracy.

**In summary, this work has the following contributions:**

1. We introduce PAAM, *an end-to-end algorithm that learns the importance scores directly from the network weights and filters*. Our method allows extracting hidden correlations in

the filter weights for training the scores, rather than relying only on the weight magnitudes. The feature maps are multiplied by our learned scores during training. This way *our method implicitly accounts for the data samples through loss propagation*, enabling PAAM to enjoy the advantages of both data-free and data-informed methods.

2. PAAM *automatically calculates global importance scores for all filters and determines layer-specific budgets* with only one global hyper-parameter.

3. We empirically investigate the pruning performance of PAAM in various pruning tasks and compare it to advanced state-of-the-art pruning algorithms. *Our method proves to be competitive and yields higher pruning ratios while preserving higher accuracy.*

## 2 PRUNING BY ACTIVE ATTENTION MANIPULATION ALGORITHM

In this section, we first introduce our notation and then incrementally describe our pruning algorithm.

**2.1 Notation.** The filter weights of layer $l$ are given by the tuple $\mathcal{F}_l \in \mathbb{R}^{F \times C \times K \times K}$ where $F$ is the number of filters, $C$ the number of input channels, and $K$ the size of the convolutional kernel. The feature maps of layer $l$ are given by the tuple $A_l \in \mathbb{R}^{F \times H \times W}$ where $H$ and $W$ are the image height and width respectively. For simplicity, we ignore the batch dimension in our formulas.

**2.2 Score Learning.** The score-learning function of PAAM, for a layer $l$ of the CNN to be pruned, can be intuitively understood as a single-layer independent network, whose inputs are the filter weights $\mathcal{F}_l$ of layer $l$, and the outputs are the scores $S_l$ associated to the filters. The network first transforms the input weights $\mathcal{F}_l$ to a score vector $\mathcal{F}_l\, W^{\mathcal{F}}$, whose length equals the number $F$ of filters of layer $l$, and then passes the result through an activation function $\phi$, properly spreading the scores within the $[0, 1 + \epsilon]$ interval. The resulting scores are then used by an $l1$ regularization term of the cost function. The choice of $\phi$ and regularization term is discussed in the next sections. Formally:

$$S_l = \phi(\mathcal{F}_l\, W^{\mathcal{F}}) \tag{1}$$

where $\phi$ is the activation function and $W^{\mathcal{F}} \in \mathbb{R}^{(F \times C \times K \times K) \times F}$ is the weight matrix. This transformation, through $W^{\mathcal{F}}$ and $\phi$ (vanilla PAM), is similar to additive attention Bahdanau et al. (2014). We will therefore refer in the following to the score-learning network as the attention network (AN). Intuitively, the AN weight matrix $W^{\mathcal{F}}$ captures the hidden correlations among filters. Unfortunately, for layers with a large number of filters, as for ImageNet, this matrix will become too large to train.

To capture the correlations in a scalable fashion, we would like to first partition the input in chunks, compute the correlations within each chunk, and then compute the correlations among chunks. But this is exactly what a scaled dot-product attention achieves Vaswani et al. (2017a). First, PAAM reformats the input as a binary matrix $\mathcal{F}_l \in \mathbb{R}^{(F) \times (C \times K \times K)}$, where the chunks are its rows. Second, it obtains the correlations within each chunk as the queries $Q_l = \mathcal{F}_l\, W^Q$ and keys $K_l = \mathcal{F}_l W^K$. Here, the query and key weight matrices $W^Q, W^K \in \mathbb{R}^{(C \times K \times K) \times d_l}$ are much smaller, as they consider only a chunk, and $d_l = F$ is the hidden dimension of the layer. The queries and keys are of shape $Q_l, K_l \in \mathbb{R}^{(F) \times (d_l)}$. Third, PAAM obtains the cross correlations among chunks by multiplying the query matrix $Q_l$ with the transpose $K_l^T$ of the key matrix. Finally, the scores are obtained by averaging and normalizing the result, and passing it through the activation function $\phi$. Formally:

$$S_l = \phi\left(\frac{mean(Q_l \times K_l^T)}{\alpha\sqrt{d_l}}\right) \tag{2}$$

where $\alpha$ is a scaling factor that we tune. Note that PAAM does not need to learn the value-weight matrix, compute the values, and multiply them with the above result, as it is only interested in the correlation among filters. Learning filter weights is left to CNN training. However, the scores $S_l$ learned by PAAM, are then pointwise ($\odot$) multiplied by the feature maps $A_l$ of the same layer:

$$A_l^{'} = S_l \odot A_l \tag{3}$$

The closer a filter score is to 1, the more the corresponding feature map is preserved. Note also that by using an analog value for scores while training the AN, allows PAAM to compare the relative importance of scores later on. In particular, this is useful when PAAM employs the globally allocated budget, and the cumulative score distribution, to select the filters to be used during CNN training.

**2.3 Activation Function.** SoftMax is the typical choice of activation function in additive attention when computing importance Vaswani et al. (2017b). However, SoftMax is not a suitable choice for filter scores. While the range of its outputs is between 0 and 1, the sum of its outputs has to be 1, meaning that either all scores will have a small value, or there will be only one score with value 1.

In contrast to SoftMax, we would intuitively want that many filter scores are possibly close to 1. More formally, the scores should have the following three main attributes: 1) All filter scores should have a positive value that ranges between 0 and 1, as is the case in SoftMax. 2) All filter scores should adapt from their initial value of 1, as we start with a completely dense model. 3) The filter-scores activation function should have non-zero gradients over their entire input domain.

Sigmoidal activations satisfy Attributes 1 and 3. However, they have difficulties with Attribute 2. For high temperatures, sigmoids behave like steps, and scores quickly converge to 0 or 1. The chance these scores change again is very low, as the gradient is close to zero at this point. Conversely, for low temperatures, scores have a hard time converging to 0 or 1. Finding the optimal temperature needs an extensive search for each of the layers separately. Finally, starting from a dense network with all scores set to 1 is not feasible.

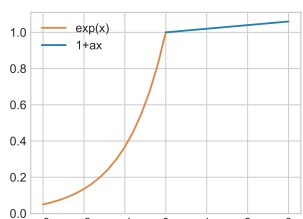

Figure 3: The leaky-exponential activation function.

To satisfy Attributes 1-3, we designed our own activation function, as shown in Figure 3. First, in order to ensure that scores are positive, we use an exponential activation function and learn its logarithm value. Second, we allow the activation to be leaky, by not saturating it at 1, as this would result in 0 gradients, and scores getting stuck at 1. Formally, our leaky-exponential activation function is defined as follows, where $a$ is a small value (an ablation study in provided in Appendix):

$$\phi(x) = \begin{cases} e^x & \text{if } x < 0 \\ 1.0 + ax & \text{if } x \geq 0 \end{cases} \tag{4}$$

**2.4 Optimization Problem.** The PAAM optimization problem can be formulated as in Equation (5), where $\mathcal{L}$ is the CNN loss function, and $f$ is the CNN function with inputs $x$, labels $y$, and parameters $\mathcal{W}$, modulated by the AN with inputs the filter weights, parameters $\mathcal{V}$, and outputs the scores $S$. Let $\|g(S_l, p)\|_1$ denote the $l_1$-norm of the binarized filter scores of layer $l$, $F_l$ the number of filters of layer $l$, $L$ the number of layers, and $p$ the pruning ratio. Then PAAM should preserve only as many filters as desired by $p$, while minimizing at the same time the loss function.

$$\min_{\mathcal{V}} \mathcal{L}(y, f(x; \mathcal{W}, S)) \qquad s.t. \ \sum_{l=0}^{L} \|g(S_l, p)\|_1 - p \sum_{l=0}^{L} F_l = 0 \tag{5}$$

Function $g(S_l, p)$ is discussed in next section. The budget constraint is addressed by adding an $l1$ regularization term to the loss function, while training the weights of the AN. This term employs the analog scores of the filters in each layer, which are computed as described above. Formally:

$$\mathcal{L}'(S) = \mathcal{L}(y, f(x; \mathcal{W}, S)) + \lambda \sum_{l=0}^{L} \|S_l\|_1 \tag{6}$$

where $\lambda$ is a global hyper-parameter controlling the regularizer's effect on the total loss. Since the loss function consists now of both classification accuracy and the $l_1$-norm cost of the scores, reducing filter scores to decrease $l_1$ cost, directly influences accuracy as scores multiply the activation maps.

In more detail, by incorporating the classification and the $l_1$-norm of the scores in the loss function, the effect of the score of each filter is accounted for in the loss value in two ways:

1. When multiplied by small scores, feature maps have less influence on classification. This may result in a larger loss if these maps are important and vice versa.

2. On the other hand, the part of the loss function consisting of the $l_1$-norm of the scores, decreases the loss value when there are more scores with small values.

In some sense, this duality mimics the way synapses are pruned or strengthened through habituation or conditioning in nature. If the use of a synapse does not lead to a reward it is pruned, otherwise, it is strengthened. Moreover, pruning is essential in order to avoid saturation and enable learning.

The value of $\lambda$ plays a very important role in keeping the balance between the two factors, and in controlling the pruning ratio. When $\lambda$ is 0, all scores will stay close to 1, and when it is large, all scores will converge to 0. During training of the AN layers, we freeze all CNN parameters. The scores are directly learned from the filter weights, reflecting attributes such as the filter-weights magnitude and the correlation among the different filters of the same layer. A globally pruned network is obtained by simultaneously training the scores of all layers. This allows propagating the scored influence of the feature maps of one layer to the feature maps and scores of the next layers.

**2.5 Training schedule.** PAAM starts training the CNN by a number of warm-up epochs. During this phase, it trains the dense CNN with all filter scores fixed to 1, and the AN weights frozen. After the warm-up phase, PAAM starts training both the AN weights and the CNN weights.

As discussed, PAAM multiplies the CNN feature maps by the AN analog scores. This way, when a score is 1, the corresponding feature map is fully preserved. As the score gets smaller, its feature map is getting weaker, since it is multiplied by a value smaller than 1, and finally almost completely pruned, as the score gets closer to 0. However, while the scores are getting trained, the network weights can learn to adapt to the feature-map intensity. This means that, although a filter may have a low score, the loss of the feature map intensity can be compensated by magnifying the feature map itself. In order to prevent this, PAAM uses an alternating training approach Peste et al. (2021).

First, PAAM trains the AN for a few epochs, while freezing all CNN parameters. Then it trains the sparse CNN for a few epochs, while freezing all AN parameters. Sparsity is obtained by multiplying feature maps $A_l$ with a binary version $g(S_l, p)$ of scores learned so far. Function $g$ is described in the next section. AN training and CNN training alternate for a predefined number of training cycles. After the training

---

**Algorithm 1** PAAM

**Inputs:** Mini-batches $\mathcal{B}$; CNN $N$ after warm-up, parametrized by $\mathcal{W}$; AN parametrized by $\mathcal{V}$; Regularization hyper-parameter $\lambda$; PAAM-training epochs $E_p$; Training mini-epochs $E_s$.

**Output:** Pruned CNN for fine-tuning.

**for** $i$ in $E_p$ **do**
  **for** $j$ in $E_s$ **do**
    **for** $b$ **in** $\mathcal{B}$ **do**
      $\min_{\mathcal{V}} \mathcal{L}(y, f(x; \mathcal{W}, S)) + \lambda \sum_{l=0}^{L} \|S_l\|_1$
      calculate gradients w.r.t $\mathcal{V}$
      Update $\mathcal{V}$ by the optimizer
    **end for**
  **end for**
  **for** $j$ in $E_s$ **do**
    **for** $b$ **in** $\mathcal{B}$ **do**
      $\min_{\mathcal{W}} \mathcal{L}(y, f(x; \mathcal{W}, S))$
      calculate gradients w.r.t $\mathcal{W}$
      Update $\mathcal{W}$ by optimizer
    **end for**
  **end for**
**end for**
**Return** $N$ with parameters $\mathcal{W}$

---

phase is completed, PAAM finishes the pruning procedure, by fine-tuning the sparse model. Note that PAAM does not need a pretrained dense model to find the importance of filters or feature maps. Even when starting from scratch, after a warm-up phase, it can find an optimal sparse sub-network by training the AN and CNN weights, respectively, as discussed above.

**2.6 Binarization function.** As mentioned above, during CNN training, PAAM fixes the scores to 1, for all filters retained according to their score magnitude and pruning ratio. All other filters have score 0. The analog-to-digital transformation is accomplished by the function $g(s, p)$, as follows:

$$g(s, p) = \begin{cases} 1 & \text{if } s \geq \theta(S, p) \\ 0 & \text{otherwise.} \end{cases} \tag{7}$$

where $s$ is the analog-score value of a filter, and $\theta(S, p)$ is the optimal pruning threshold that is computed, by considering the cumulative distribution of all scores $s \in S$, and the pruning ratio $p$. We will discuss how PAAM has computed $\theta(S, p)$ for Cifar10 and ImageNet, in Section 3.

**2.7 Complete Algorithm** Algorithm 1 describes PAAM's intermediate stage. While training the AN parameters, it uses Equations (4-6). While training the CNN parameters, it uses Equation (7).

## 3 EXPERIMENTAL EVALUATION

In this section, we first present our implementation details and then discuss our experimental results.

**Baselines.** We compare PAAM to numerous standard and advanced pruning methods. The models include: L1 norm Li et al. (2017), neuron-importance score propagation (NISP) Yu et al. (2017), soft

Table 1: Results on the Cifar10 dataset with ResNet56 and ResNet110 for medium (top part) and for large (bottom part) pruning ratios, respectively. The quantity $\Delta$ shows the difference in the accuracy (Acc) between the baseline dense model used, and the resulting pruned network. Note that PAAM does not use a pretrained baseline. Numbers are taken from the reported results of the cited papers.

| | ResNet56 | | | | |
| Method | Baseline Acc(%) | Pruned Acc(%) | $\Delta$(%) | ↓Params(%) | ↓Flops(%) |
| --- | --- | --- | --- | --- | --- |
| $l1$ Norm (2017) Li et al. (2017) | 93.04 | 93.06 | +0.02 | 13.7 | 27.6 |
| NISP (2017) Yu et al. (2017) | N/A | N/A | -0.03 | 42.2 | 35.5 |
| SFP (2018) He et al. (2018a) | 93.59 | 93.78 | +0.19 | N/A | 41.1 |
| DCP (2018) Zhuang et al. (2018) | 93.80 | 93.59 | -0.31 | N/A | 50.0 |
| DCP-Adapt (2018) Zhuang et al. (2018) | 93.80 | 93.81 | +0.01 | N/A | 47.0 |
| CCP (2019) Peng et al. (2019) | 93.50 | 93.46 | -0.04 | N/A | 47.0 |
| GAL (2019) Lin et al. (2019) | 93.26 | 93.38 | +0.12 | 11.8 | 37.6 |
| HRank (2020) Lin et al. (2020) | 93.26 | 93.52 | +0.26 | 16.8 | 29.3 |
| DMC (2020) Gao et al. (2020) | 93.62 | 93.69 | +0.07 | N/A | 50.0 |
| NPPM (2021) Gao et al. (2021) | 93.04 | 93.40 | +0.36 | N/A | 50.0 |
| CHIP (2021) Sui et al. (2021) | 93.26 | 94.16 | +0.90 | 42.8 | 47.4 |
| **PAAM(Vanilla)** | | **94.28** | **+1.02** | 52.3 | 49.3 |
| **PAAM(KQ)** | | 94.01 | +0.75 | **54.0** | **52.3** |
| AMC (2018) He et al. (2018d) | 92.80 | 91.90 | -0.90 | N/A | 50.0 |
| GAL (2019) Lin et al. (2019) | 93.26 | 91.58 | -1.68 | 65.9 | 60.2 |
| HRank (2020) Lin et al. (2020) | 93.26 | 90.72 | -2.54 | 68.1 | 74.1 |
| CHIP (2021) Sui et al. (2021) | 93.26 | 92.05 | -1.21 | 71.8 | 72.3 |
| **PAAM(Vanilla)** | | 92.42 | -0.84 | **83.0** | 70.5 |
| **PAAM(KQ)** | | **93.10** | **-0.16** | 70.0 | **74.7** |
| | ResNet110 | | | | |
| $l1$ Norm (2017) Li et al. (2017) | 93.53 | 93.30 | -0.23 | 32.4 | 38.7 |
| NISP (2018) Yu et al. (2017) | N/A | N/A | -0.18 | 43.78 | 43.25 |
| SFP (2018) He et al. (2018a) | 93.68 | 93.86 | +0.18 | N/A | 40.8 |
| GAL (2019) Lin et al. (2019) | 93.50 | 93.59 | +0.09 | 18.7 | 4.1 |
| HRank (2020) Lin et al. (2020) | 93.50 | 94.23 | +0.73 | 39.4 | 41.2 |
| CHIP (2021) Sui et al. (2021) | 93.50 | 94.44 | +0.94 | 48.3 | **52.1** |
| **PAAM(Vanilla)** | | **94.69** | **+1.19** | **54.9** | 51.3 |
| **PAAM(KQ)** | | 94.63 | +1.13 | 48.3 | 40.9 |
| GAL (2019) Lin et al. (2019) | 93.50 | 92.74 | -0.76 | 44.8 | 48.5 |
| HRank (2020) Lin et al. (2020) | 93.50 | 92.65 | -0.85 | 68.7 | 68.6 |
| CHIP (2021) Sui et al. (2021) | 93.63 | 93.63 | +0.13 | 68.3 | 71.3 |
| **PAAM(Vanilla)** | | 93.68 | +0.18 | **79.3** | 74.8 |
| **PAAM(KQ)** | | **93.99** | **+0.49** | 66.5 | **77.2** |

filter pruning (SFP) He et al. (2018a), discrimination-aware channel pruning (DCP) Zhuang et al. (2018), DCP-adapt Zhuang et al. (2018), collaborative channel pruning (CCP) Peng et al. (2019), generative adversarial learning (GAL) Lin et al. (2019), filter-pruning using high-rank feature maps (HRank) Lin et al. (2020), discrete model compression (DMC) Gao et al. (2020), network pruning via performance maximization (NPPM) Gao et al. (2021), channel independence-based pruning (CHIP) Sui et al. (2021), and auto-ML for model compression (AMC) He et al. (2018d).

## 3.1 IMPLEMENTATION DETAILS

**Training Procedure.** Our first experiments are on CIFAR-10 with two different models: ResNet56 and ResNet110. For both models, PAAM does not use a fully trained network as the baseline to prune. We train each model for 50 warm-up epochs. During warm-up (see Figure 1), we use a batch size of 256 and stochastic gradient descent (SGD) as optimizer, with 0.1 as the initial learning rate, 0.9 as momentum, and 0.0005 for the weight decay. We then use PAAM to train the scores.

After warm-up, PAAM trains the AN and the CNN in alternation, with the other-network parameters frozen. PAAM trains the AN weights for 3 epochs, and uses the regularized loss defined in Equation (10). Then PAAM switches to training the CNN weights for 6 epochs, and uses the classification loss. In this phase, PAAM employs Equation equation 7 as the analog-to-digital score-conversion function. When PAAM trains the AN, the CNN feature maps get multiplied by the analog scores. PAAM repeats these two phases for 10 times, summing up to training for 90 epochs. PAAM uses the ADAM optimizer with learning rates of $10^{-6}$ / $10^{-3}$ for training the AN / CNN parameters.

After training, PAAM fine-tunes the CNN. In this stage, it removes the AN, and keeps only the CNN filters with the binary score of one. PAAM uses Equation equation 7 to compute the binary scores from their analog counterparts, which is in fact PAAM's ultimate goal in pruning. Each feature map is either removed entirely (having score zero), or is completely preserved (having score one). PAAM uses the SGD optimizer, with the same parameters as in warm-up, and tunes the CNN for 300 epochs.

Table 2: Experimental results on ImageNet with ResNet50. $\Delta$ is the difference in accuracy between baseline and pruned networks. The numbers are taken from the reported results in the cited papers.

| | Top1 Acc(%) | | | Top5 Acc(%) | | | $\downarrow$(%) | |
| --- | --- | --- | --- | --- | --- | --- | --- | --- |
| Method | Baseline | Pruned | $\Delta$ | Baseline | Pruned | $\Delta$ | Params | Flops |
| SFP (2018) He et al. (2018a) | 76.15 | 74.61 | -1.54 | 92.87 | 92.06 | -0.81 | N/A | 41.8 |
| DCP (2018) Zhuang et al. (2018) | 76.01 | 74.95 | -1.06 | 92.93 | 92.32 | -0.61 | 50.9 | 55.6 |
| FPGM (2019) He et al. (2018c) | 76.15 | 75.59 | -0.56 | 92.87 | 92.63 | -0.24 | 37.5 | 42.2 |
| CCP (2019) Peng et al. (2019) | 76.15 | 75.5 | -0.65 | 92.87 | 92.62 | -0.25 | N/A | 48.8 |
| GAL (2019) Lin et al. (2019) | 76.15 | 71.95 | -4.2 | 92.87 | 90.94 | -1.93 | 16.9 | 43.0 |
| PFP (2020) Liebenwein et al. (2020) | 76.13 | 75.21 | -0.92 | 92.87 | 92.43 | 0.44 | 30.1 | 44.0 |
| AutoPruner (2020) Luo & Wu (2020) | 76.15 | 74.76 | -1.39 | 92.87 | 92.15 | -0.72 | N/A | 48.7 |
| HRank (2020) Lin et al. (2020) | 76.15 | 74.98 | -1.17 | 92.87 | 92.33 | -0.54 | 36.6 | 43.7 |
| SCOP (2020) Tang et al. (2020) | 76.15 | 75.95 | -0.20 | 92.87 | 92.79 | -0.08 | 42.8 | 45.3 |
| DMC (2020) Gao et al. (2020) | 76.15 | 75.35 | -0.80 | 92.87 | 92.49 | -0.38 | N/A | 55.0 |
| NPPM (2021) Gao et al. (2021) | 76.15 | 75.96 | -0.19 | 92.87 | 92.75 | -0.12 | N/A | 56.0 |
| CHIP (2021) Sui et al. (2021) | 76.15 | 76.30 | +0.15 | 92.87 | 92.91 | +0.04 | 40.8 | 44.8 |
| **PAAM(KQ)** | 76.15 | **77.21** | **+1.06** | 92.87 | **93.55** | **+0.68** | **51.1** | 31.9 |

Our second experiments are on ImageNet with ResNet50. As ResNet50 has many layers with a very large number of filters, PAAM (Vanilla) runs out of memory, whereas PAAM (KQ) has no problems to scale up. In this experiment, PAAM starts with the pretrained ResNet50 model, in order to save time. PAAM alternates the training of the AN/CNN networks for 54 epochs, with 3 epochs for the AN and 6 epochs for the CNN in each iteration. PAAM then fine tunes the pruned CNN for 200 epochs. PAAM sets the batch size, momentum, weight decay and initial learning rate to 128, 0.875, 2e-05 and 0.5, respectively, and uses a cosine-annealing learning-rate scheduler with 5 as warm-up.

**Balancing the Pruned Parameters and Flops.** PAAM uses while training the AN, the regularized loss to guide the scores to the desired filter budget. However, the number of parameters of each CNN layer is different from the number of computation flops required for that CNN layer:

$$(Params)_l = C_l \times F_l \times K_l \times K_l, \quad (Flops)_l = C_l \times F_l \times K_l \times K_l \times W_l \times H_l \quad (8)$$

As the number of flops also depends on the image size, early convolutional layers, before the max-pooling layers, require more flops as the image size is larger. To properly balance pruning vs flops, we multiply the $l_1$ norm of each layer by the fraction of the image sizes of that layer and the last layer.

For experiments on ResNet56, PAAM used $\lambda = 5 \times 10^{-4}$ and $\lambda = 15 \times 10^{-4}$ for pruning ratios of 52.3% and 83.0% respectively. On ResNet110, PAAM used $\lambda = 2 \times 10^{-4}$ and $\lambda = 5.5 \times 10^{-4}$ to prune 54.9% and 79.3% of the network parameters. On ResNet50, PAAM used $\lambda = 10^{-6}$.

## 3.2 PERFORMANCE EVALUATION

**Experimental results.** Table 1 compares the performance of PAAM (Vanilla) and PAAM (KQ) to the sate-of-the-art filter-pruning methods (SOA) on Cifar10. Both versions of PAAM outperform SOA, achieving higher accuracy while pruning more parameters. Specifically, on ResNet56 PAAM results in an accuracy increase, even compared to the dense baselines, while significantly pruning parameters and flops. The same is true for ResNet110. For high pruning ratios, PAAM outperforms CHIP Sui et al. (2021), the next best method, with better accuracy while pruning more parameters and flops on ResNet56. Similarly, on ResNet110, PAAM outperforms CHIP in both accuracy and pruning.

Table 2 compares the performance of PAAM (KQ) to the SOA on ImagNet. Starting from a baseline of 76.15% accuracy, PAAM achieves a higher accuracy of 77.21% even when compared to the baseline (an increase of 1.06%), while pruning 51.1% of the parameters and 31.9% of the flops.

**Per-layer Budget Discovery.** A remarkable feature of PAAM it that it finds the optimal sparse sub-networks in a fully-automated pipeline and does not require a budget-allocation schedule per layer. Figure 4 shows the discovered networks in our experiments from ResNet56 and ResNet110.

In each residual block of the sub-networks learned by PAAM, the first layer has lower number of filters remaining after pruning, compared to the second layer. This structure is similar to the bottle-neck architecture used in ResNets with a large number of layers. It enables the network to concentrate on the most important features with less capacity, which is exactly what we are looking for with pruning. Many existing pruning algorithms use similar bottle-neck structures, when manually

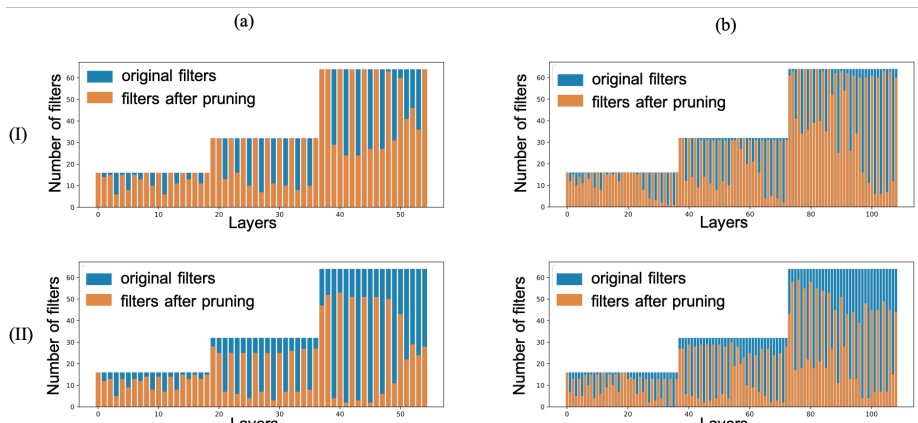

Figure 4: CNNs discovered by PAAM from ResNet56 (a) and ResNet110 (b), with medium (I) and high (II) pruning ratios. PAAM finds the optimal per-layer budgets automatically.

defining layer budgets for pruning Sui et al. (2021); Lin et al. (2020). PAAM is able to discover this pattern automatically, without supervision. Moreover, for high pruning ratios on ResNet110, there are blocks emerging with 0 remaining filters in the first layers. This shows that PAAM can also remove entire layers when required, having more freedom in the possible sparse sub-networks search space.

**The threshold value.** PAAM computes the optimal threshold $\theta(S, p)$ of Equation (7), based on the pruning ratio $p$, and the cumulative distribution of filter scores, as shown in Figure 5, for the Cifar10 experiments on ResNet56 and ResNet110. This resulted in $\theta(S, p) = 0.5$.

As one can see, the leaky-exponential activation function and the regularized-loss functions of PAAM, ensure that the majority of the scores are pushed to values close to zero and one. Hence, most of the scores lie very close to the two ends of the $[0, 1]$ interval, and $0.5$ is in its center. Moreover, as it is shown in Figure 5, the cumulative-distribution plot of scores is almost horizontal before $0.5$.

Table 3: Ablation results on Cifar10 dataset with ResNet56 pruned with different threshold values.

| Threshold($\theta$) | ↓(%)Params | ↓(%)Flops | Pruned Accuracy(%) |
|---|---|---|---|
| 0.2 | 46.0 | 36.7 | 94.30 |
| 0.3 | 50.2 | 42.4 | 94.32 |
| 0.4 | 51.6 | 46.6 | 94.29 |
| 0.5 | 52.3 | 49.3 | 94.28 |
| 0.6 | 58.2 | 57.6 | 93.97 |
| 0.7 | 64.7 | 65.9 | 93.66 |

In order to provide more insight on the proper choice for the value of the threshold, we conducted an ablation experiment on Cifar10 with the ResNet56 network. After the scores training epochs, we pruned the filters with different thresholds. This process is then followed by fine-tuning epochs. Table 3 shows the accuracies of the pruned network with each threshold value. As the table shows, the pruned parameters and flops percentages change slowly with changing the score threshold. As one can see, a threshold $\theta$ of 0.5 ensures best accuracy for most pruning of parameters and flops.

# 4 RELATED WORK

**Attention.** In recent years, attention has achieved great success in many computer-vision tasks Guo et al. (2022), from attentional CNNs to transformers Dosovitskiy et al. (2020). These works mainly focus on improving the accuracy of vision networks by learning to what (channel) or where (spatial) to pay attention to. The squeeze-and-excitation network Hu et al. (2017) is one of the pioneers in channel-attention networks which calculate channel scores from feature maps to enhance classification.

*PAAM.* To the best of our knowledge, this paper is the first to employ attention, tailored to computing filter-importance scores from filter weights, for pruning CNNs. PAAM (KQ) only uses the key and the query matrices to compute the filter correlations within a layer, and disregards the value matrices.

**Pruning.** Filter pruning is a very popular structured-pruning method for sparsifying CNNs, which supports storage reduction and processing efficiency, without requiring any special library. One can roughly classify existing filter-pruning methods into three main categories, based on their filter-selection approaches: data-free, data-informed, and training-aware methods, respectively.

*Data-free filter pruning.* Following-up on the weights-magnitude-pruning method, where the weights with the smallest absolute values are considered the least important, Li et al. (2017) uses the sum of

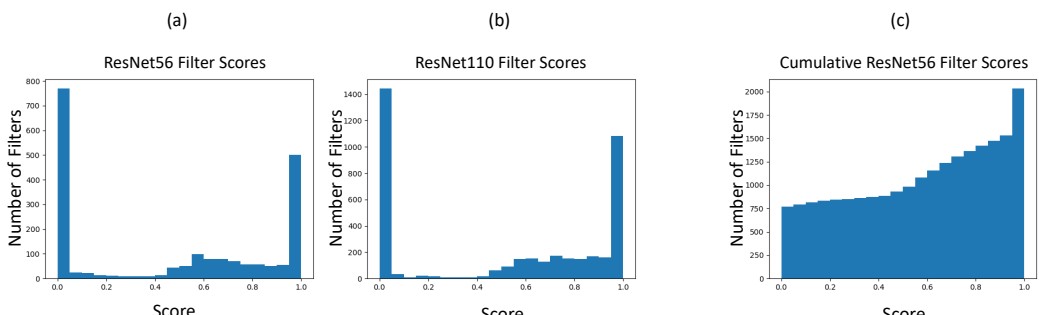

Figure 5: Filter-score histograms for Cifar10 experiments. Most filters have zero and one scores.

absolute values of the weights in a filter (the $l1$ norm) to prune the filters with the smallest weight values. He et al. (2018a) dynamically prunes the filters with the smallest $l2$ norm in each epoch by setting them to zero and repeating this process in each epoch during training. He et al. (2019) uses the geometric median of filters as the pruning criterion. In summary, although data-free methods can gain acceptable performance levels, several works have later shown that considering the training data during the pruning process, will notably improve pruning precision Hoefler et al. (2021).

*Data-informed filter pruning.* Many pruning methods focus on feature maps as they provide rich information from both data distribution and filters. Lin et al. (2020) prunes filters whose feature maps have lowest ranks, and Liebenwein et al. (2020) uses a sensitivity measure to prune filters with lowest effect on the outputs, giving provable sparsity guaranties. Motivated by the importance of inter-channel perspective for pruning, Sui et al. (2021) uses the nuclear norm of feature maps as an independence metric to prune the filters whose feature maps are the most dependent on the others.

*Training-aware filter pruning.* These methods use training to learn a filter-importance metric or guide the network to a sparse structure. Exploiting magnitude pruning, some add regularization factors to the loss, to guide filters values towards zero. Wen et al. (2016) and Louizos et al. (2018) use group-lasso and $l0$ regularization, respectively. Instead of solely relying on the weight magnitudes, Zhuang et al. (2018) proposes a discrimination-aware channel-pruning method, by defining per-layer classification losses. Gao et al. (2020) trains binary gate functions with straight-through estimators and Gao et al. (2021) focuses on training binary gates, by directly maximizing the accuracy of subnetworks.

*PAAM.* To the best of our knowledge, none of the above methods, is able to automatically learn the filter-importance scores from the filter weights, extract hidden correlations among filters, automatically calculate global importance scores for all filters, and determine layer-specific budgets, all at the same time during training, and thus taking advantage of both data-free and data-informed methods.

## 5 CONCLUSION

We proposed a novel, end-to-end, attention-based, filter-pruning algorithm called PAAM (pruning by active attention manipulation). PAAM learns the filter-importance scores, through gradient descent on the CNN equipped with a filter-attention network (AN), and a classification loss-function regularized with the $l1$ norm of the filter scores. The AN computes the filter scores from the filter weights and automatically finds the correlations among filters within each layer. The $l1$ regularization, encourages the decrease of scores and finds the filter correlations across layers. In contrast to a large spectrum of advanced pruning algorithms, PAAM does not necessarily require a pretrained baseline CNN to prune from. It can rather sparsify dense networks from scratch, through cycles of gradient descent.

We showed on a comprehensive set of experiments on Cifar10 and ImageNet, that much better compression rates are achievable through the use of PAAM for ResNet50, ResNet56, and ResNet110, while even surpassing baseline accuracy. PAAM is able to compute the global filter-importance scores and to automatically associate pruning budgets to CNN layers, without layer-specific hyperparameters. In this way, PAAM is taking advantage of both the data-free and the data-informed pruning methods.

It is our hope that future work is going to start using PAAM in resource-hungry applications domains, such as those of neural-architecture search Mellor et al. (2021). The application of PAAM promises to also result in compressed neural networks endowed with salient features, such as accuracy and small compute footprint, automatically distilled during training for resource-constrained environments.

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

# A  APPENDIX

## A.1  PAAM'S EFFECT ON ITERATIVE INFERENCE IN RESNETS

In this section, we proceed with an analysis of the effect that PAAM has on the iterative feature-refinement process in residual networks Greff et al. (2016).

ResNets He et al. (2016) are known to enhance representation learning in deeper layers via an iterative feature-refinement scheme Greff et al. (2016). This scheme suggests that input features to a ResNet do not create new representations. Rather, they gradually and iteratively refine the learned features of the initial residual blocks Jastrzebski et al. (2018). Iterative refinement of features was shown to be necessary for obtaining attractive levels of performance, while their disruption hurts performance.

As AN modifies the underlying model structure and the feature maps of a residual block, it is very important to investigate if PAAM preserves the iterative feature-refinement property of ResNets.

To make this analysis precise, let us first formalize iterative inference as discussed in Jastrzebski et al. (2018): A residual block $i$ in a ResNet with $M$ blocks, performs for the input feature $\mathbf{x}_i$ the following transformation: $\mathbf{x}_{i+1} = \mathbf{x}_i + f_i(\mathbf{x}_i)$. Hence, the following loss function $\mathcal{L}$ can be recursively applied to the network Jastrzebski et al. (2018):

$$\mathcal{L}(\mathbf{x}_M) = \mathcal{L}(\mathbf{x}_{M-1} + f_{M-1}(\mathbf{x}_{M-1})). \tag{9}$$

A first-order Taylor expansion of this loss, while ensuring that $f_j$'s magnitude is small, is a good approximation to formally investigating the iterative inference Jastrzebski et al. (2018). Thus:

$$\mathcal{L}(\mathbf{x}_M) = \mathcal{L}(\mathbf{x}_i) + \sum_{j=i}^{M-1} f_j(\mathbf{x}_j) . \frac{\partial \mathcal{L}(\mathbf{x}_j)}{\mathbf{x}_j} + \mathcal{O}(f_j^2(\mathbf{x}_j)). \tag{10}$$

This approximation reveals that the $i$-th residual block, modifies features $\mathbf{x}_i$ with roughly the same amount of $f_i(\mathbf{x}_i)$ as that of $\frac{\partial \mathcal{L}(\mathbf{x}_i)}{\mathbf{x}_i}$. This implies a moderate reduction of loss as we transition from the $i$-th to the $M$-th block, as an iterative refinement scheme Greff et al. (2016); Jastrzebski et al. (2018). The refinement step for a vanilla residual block can be computed by the squared norm of $f_i(\mathbf{x}_i)$, and can be normalized to the input feature as: $\|f_i(\mathbf{x}_i)\|_2^2 / \|\mathbf{x}_i\|_2^2$.

Any modification to the structure of the residual network (e.g., in PAAM) causes a change in the refinement step. This step has to be investigated if it does or it does not modify the iterative inference.

**Lemma A.1.** *The iterative feature-refinement scale is bounded for ResNets with PAAM as follows, with parameter $\epsilon$ from the leaky integrator and $0 < \delta \leq 1 + \epsilon$:*

$$\frac{\delta}{1+\epsilon} \frac{\|f_i(\boldsymbol{x}_i)\|_2^2}{\|\boldsymbol{x}_i\|_2^2} \leq \frac{\|S_i \odot f_i(\boldsymbol{x}_i)\|_2^2}{\|S_i \odot \boldsymbol{x}_i\|_2^2} \leq \frac{1+\epsilon}{\delta} \frac{\|f_i(\boldsymbol{x}_i)\|_2^2}{\|\boldsymbol{x}_i\|_2^2} \tag{11}$$

*Proof.* A ResNet block $i$ that is equipped with an SbF-Pruner layer, transforms the features $\mathbf{x}_i$ with the following expression: $\mathbf{x}_{i+1} = S_i \odot (\mathbf{x}_i + f_i(\mathbf{x}_i))$, where $S_i$ stands for the score vector computed by PAAM. The refinement step is given by $\|S_i \odot f_i(\mathbf{x}_i)\|_2^2$ and its input-normalized representation is $\|S_i \odot f_i(\mathbf{x}_i)\|_2^2 / \|S_i \odot \mathbf{x}_i\|_2^2$.

*Deriving the upper bound:* Assuming that every element in $S_i$ is between $\delta$ and $1+\epsilon$, for $0 < \delta \leq 1+\epsilon$, we have:

$$\|S_i \odot f_i(\mathbf{x}_i)\| \leq \|S_i\|_\infty \|f_i(\mathbf{x}_i)\| \leq (1+\epsilon) \|f_i(\mathbf{x}_i)\| \tag{12}$$
$$\|S_i \odot \mathbf{x}_i\| \geq \|S_i\|_{min} \|\mathbf{x}_i\| \geq \delta \|\mathbf{x}_i\|. \tag{13}$$

As a consequence, the following upper-bound inequality holds for the iterative inference:

$$\frac{\|S_i \odot f_i(\mathbf{x}_i)\|_2^2}{\|S_i \odot \mathbf{x}_i\|_2^2} \leq \frac{1+\epsilon}{\delta} \frac{\|f_i(\mathbf{x}_i)\|_2^2}{\|\mathbf{x}_i\|_2^2} \tag{14}$$

*Deriving the lower bound:* Assuming that every element in $S_i$ is between $\delta$ and $1 + \epsilon$, we have:

$$\|S_i \odot f_i(\mathbf{x}_i)\| \geq \delta \|f_i(\mathbf{x}_i)\| \tag{15}$$

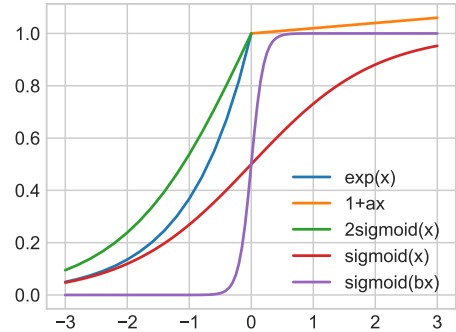

Figure 6: Activation functions.

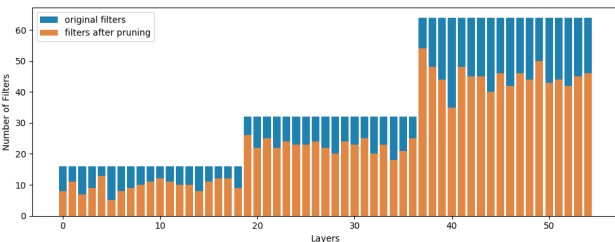

Figure 7: Network discovered by PAAM with Sigmoid activation function.

$$\|S_i \odot \mathbf{x}_i\| \leq (1 + \epsilon) \|\mathbf{x}_i\|. \tag{16}$$

As a consequence, the following lower-bound inequality holds for the iterative inference:

$$\frac{\|S_i \odot f_i(\mathbf{x}_i)\|_2^2}{\|S_i \odot \mathbf{x}_i\|_2^2} \geq \frac{\delta}{1 + \epsilon} \frac{\|f_i(\mathbf{x}_i)\|_2^2}{\|\mathbf{x}_i\|_2^2} \tag{17}$$

Inequalities equation 14 and equation 17 prove the the stated lemma. $\qquad\square$

Lemma A.1 has profound implications in practice. It indicates that the iterative-inference property of the ResNets equipped with PAAM is both lower and upper bounded. These ResNets not only get compressed in size, but also preserve the representation learning capabilities of ResNets between these two bounds. The bounds themselves can be fine tuned with the parameter $\lambda$ of Equation equation 6.

### A.2 ABLATION ON ACTIVATION FUNCTIONS

In this section, we provide additional experiments on the choice of the activation function for PAAM layers. We compare the performance of the vanilla PAAM on CIFAR10 with ResNet56 with four different activation functions presented in Figure 6.

We start with the Sigmoid function. As we already mentioned in the paper, Sigmoid is a favourable function, satisfying the first and third conditions we look for in our activation function. In our experiment with Sigmoid, the scores did not converge to values close to 0 and 1 and mostly stayed at values close to 0.5, making it hard to control the pruning ratio by the loss regularization factor. We further tried the Sigmoid function with higher temperatures, $\sigma(bx)$ with $b = 100$. In this experiment, the scores converged to 0 and 1. As discussed in the manuscript, high temperature helps the Sigmoid to converge. But the scores converge at a fast rate to 0 or 1, and hardly change in future epochs, resulting in a uniform random pruning budget allocation scenario. Figure 7 shows the network discovered by PAAM with high temperature Sigmoid activation function. Compare this to the layer specific pruning results shown in the manuscript in Figure 4(a). Table 4 shows that high temperature Sigmoid activation function results in poor accuracy for the pruned network. In order to be able to use Sigmoid as activation function, extensive per-layer tuning for the temperature is required.

To improve the results with Sigmoid, we experimented with an activation function similar to our leaky-expo, which we call leaky-2sigmoid. leaky-2sigmoid is initialized by ones with output range in $[0, 2]$ and has an affine linear component for outputs greater than 1 as follows:

$$\phi(x) = \begin{cases} 2\sigma(x) & \text{if } x < 0 \\ 1.0 + ax & \text{if } x \geq 0 \end{cases} \tag{18}$$

As one can see in Figure 6, this function is very similar to leaky-expo. The results shown in Table 1 illustrate that PAAM with leaky-2sigmoid and leaky-expo perform competitively, with leaky-expo resulting in better compression rate. This indicates that starting the scores from 1, which is starting from a fully dense network, plays an important role in the results obtained by PAAM.

Table 4: Ablation study with different activations

| Activation-Function | Pruned Acc(%) | ↓Parameters(%) |
|---|---|---|
| Sigmoid(x) | not converged | not converged |
| Sigmoid(bx) | 89.68 | 51.5 |
| Leaky-2Sigmoid | 94.10 | 48.2 |
| Leaky-expo | **94.28** | **52.3** |

### A.3 EXPERIMENTAL DETAILS

In this section, we provide the details of the experiments conducted on Cifar10 and ImageNet datasets. Table 5 shows the training epochs and $\lambda$ value for the scores loss regularizer for our experiments. In all experiments, during pruning epochs, we train the PAAM layers for 3 epochs and then switch to training the network weights for 6 epochs and repeat this set of 9 epochs.

Table 5: Hyperparameters used in Cifar10 experiments.

| Model | Warm-up | Pruning | Fine-tuning | $\lambda$ | ↓Params(%) | ↓Flops(%) |
|---|---|---|---|---|---|---|
| ResNet56 | 50 | 90 | 300 | $5e^{-4}$ | 52.3 | 49.3 |
| | | | | $1.5e^{-3}$ | 83.0 | 70.5 |
| ResNet110 | 50 | 90 | 300 | $2e^{-4}$ | 54.9 | 51.3 |
| | | | | $5.5e^{-4}$ | 79.3 | 74.8 |
| ResNet50 | pre-trained | 54 | 200 | $1e^{-6}$ | 51.1 | 23.1 |

**Cifar10.** We used Vanilla and KQ PAAM for Cifar10 datasets. We didn't use fully pre-trained networks, but rather did 50 warm-up epochs before pruning.

**ImageNet.** We used KQ-PAAM for ImageNet experiments. We set the hidden dimension $d_l$ of each layer to number of filters of that layer, devided by the layer expansion parameter, which is 4 in ResNet50. We normalised the scaling factor by $1/\alpha$, set to the sum-of-squares root of the number of filters in the last layers of each block. We used the pre-trained pytorch model for ResNet50. For pruning and fine-tuning we augment the data by a random-resized crop, random-horizontal flip, and trivial augmentation. We also normalized the data samples. For pruning epochs, we use Adam, and set the learning rates to $10^{-6}$ and $10^{-3}$ for score training and network training phases, respectively.

