# OpenReview forum: "Pruning by Active Attention Manipulation"
_ICLR.cc/2023/Conference — Submitted to ICLR 2023_

### Official Review · Reviewer_wYHx · 2022-10-25

**Confidence:** 3
**Correctness:** 3
**Technical Novelty And Significance:** 2
**Empirical Novelty And Significance:** 3
**Recommendation:** 6

**Clarity, Quality, Novelty And Reproducibility:**

- The main idea using attention mechanisms for filter pruning is intuitive and well-motivated.
- Although existing attention mechanisms are used, interesting elements (e.g., nonlinear activation, alternating training procedure) are introduced.
- The effectiveness of the method is well supported by extensive experiments.
- The implementation and reproducibility concern me a little bit, but I believe this can be relieved if the authors release their codes.
- Typo: 7p sate-of-the-art -> state-of-the-art


**Strength And Weaknesses:**

Strength
- The idea that utilizes attention mechanisms to compute the importance scores of filters while considering the correlations between them is well-motivated.
- Although the authors used the existing attention mechanisms of previous literature, they introduce several interesting elements to properly exploit them for structured pruning.
- I think the experimental validation is quite good. The baselines are strong, and the results of the proposed method are either SOTA or competitive.
- I appreciate the visualization of Figure 4 that shows the selected filters after pruning and the related discussions.
- The paper is well backed-up by a comprehensive supplementary material that contains the in-depth discussion and ablation study on the proposed method as well as the details required for reproducibility.
- The paper is generally quite thorough and generally feels complete: I believe it is generally ready for publication if it is decided by the reviewers that significance is sufficient to warrant publication.

Weakness
- The proposed method was validated with only ResNets. Is this method also effective for small-size or other networks, including MobileNets and VGG? It would be helpful to present additional results with other architectures.
- I think this paper includes only the performance after finetuning, and it is difficult for me to identify whether good accuracy comes from the power of finetuning or the effectiveness of PAAM. It would be good to add the performance before finetuning (but after channel pruning) and compare it with at least one baseline method.
- My major concern is that the proposed method requires numerous hyperparameters including the number of warm-up epochs, the number of PAAM-training epochs, the number of alternating training epochs, the regularization weighting, and different optimizer schedules for AN/CNN parameters. How did the authors set these values? Although the paper includes quite detailed numbers, it would be very helpful to provide some guidelines to determine hyperparameters.
- I would highly recommend the authors to make their codes publicly available for the research community. Implementing the proposed method with attention networks seems not to be simple, and releasing the codes would be very helpful for reproduction and future studies.


**Summary Of The Paper:**

This paper proposes to compute the importance of filters with attention weights. Although the authors utilize existing attention mechanisms (i.e., additive attention and scaled dot-product attention), they properly design a nonlinear activation that considers the range of values and gradients, an alternating training process of model parameters and attention weights, a histogram-based analysis to determine the pruning threshold. Experiments on CIFAR and ImageNet with ResNets show the superiority of the proposed method.

**Summary Of The Review:**

In general, I believe this work did a good job in terms of applying attention mechanisms for structured pruning. Although there exists a concern regarding the reproducibility, I believe it can be resolved if the authors can release their codes. The experimental results and analyses are quite good and thorough.

---

> ### Author Response · Authors · 2022-11-18
> **Clarifications and New Experiments**
>
> We would like to thank the reviewer for their positive and constructive evaluation of our work. Below, we provide an answer to each identified weakness by the reviewer.
>
> **Code:**
> We have submitted our code in full at the time of submission in the supplementary materials. We will also make sure to release the code in a public repository upon acceptance of our paper.
>
> **Other networks:**
> As suggested by the reviewer, we performed a new experiment by PAAM on one other network. We pruned VGG16 by PAAM(KQ) on the Cifar10 dataset. For this experiment, we used the same hyperparameters as our other experiments on Cifar10 and used 0.0005 for the lambda value. The table below shows the results of PAAM compared to other methods. As the results show, PAAM can prune 92.5% of the parameters in VGG16 which is 10.9% higher than CHIP while achieving higher accuracy.
>
> |    Method      | Baseline Acc | Pruned Acc | ∆ | Params | Flops |
> |----------|--------------|------------|--------|--------|-------|
> | L1 norm | 93.96        | 93.40      | -0.56  | 34.3   | 64.0  |
> | GAL      | 93.96        | 93.77      | -0.19  | 77.6   | 39.6  |
> | HRank    | 93.96        | 93.43      | -0.53  | 82.9   | 53.5  |
> | CHIP     | 93.96        | 93.86      | -0.10  | 81.6   | 58.1  |
> | PAAM     |              | **93.93**      | **-0.03**  | **92.5**   | **58.9**  |
>
> **Fine-tuning:**
> Finetuning is an important stage in all pruning methods. For instance, in Cifar10 experiments, CHIP uses a network pre-trained for 200 epochs, calculates the channel importance scores, and then fine-tunes the model for 300 epochs, which results in a total of **500** epochs. NPPM uses a network pre-trained for 200 epochs, trains the channel gates for 200 epochs, and then fine-tunes the model for 200 epochs, which makes it into a total of **600** epochs. Similarly, DMC uses a network pre-trained for 200 epochs, trains the channel gates for 300 epochs, and then fine-tunes the model for 160 epochs, which results in a total of **660** epochs. For the same experiments, PAAM starts with 50 warm-up epochs, followed by alternating training for 90 epochs, and then fine-tuning for 300 epochs, which makes it a total of **440** epochs.
>
> While fine-tuning plays an important role in improving performance, it is not the main cause of the high accuracy of PAAM compared to other methods. In order to better show this, we repeated our experiments with a smaller number of fine-tuning epochs. We start with 0 epochs, meaning no fine-tuning. In this case, we are training the model for only 50+90=140 epochs, which is even less than the 200 epochs needed to pre-train the base model. We then use 60 epochs to train the model for 200 epochs in total which brought PAAM to an accuracy of 93.44 which is already higher than the dense base model used in CHIP (with an accuracy of 93.26).
>
> Below is the table for the following numbers of fine-tuning epochs: 0, 60, 100, 200, and 300.
>
> | epochs   | 0     | 60    | 100   | 200   | 300   |
> |----------|-------|-------|-------|-------|-------|
> | Accuracy | 90.71 | 93.44 | 93.83 | 94.01 | 94.28 |
>
> **Hyper-parameters:**
> We did not perform any hyper-parameter tuning except for lambda as described below. We used 300 epochs for fine-tuning similar to CHIP. The number of epochs for the warm-up and alternating tuning was inspired by [1]. We used the same values for these hyper-parameters in all of our experiments. However, one could achieve even better results by searching for the optimum number of epochs for each phase.
>
> For the hyper-parameter lambda, we performed a grid search to gain the proper pruning ratios, since lambda is the only variable that controls the sparsity of the network. The higher the value of lambda is, the more parameters and flops are pruned.
>
> For the CNN learning rate, we did not perform hyper-parameter tuning and set it to 0.001. We used a lower learning rate for the AN because we were interested in a low convergence speed for the scores.

---

### Official Review · Reviewer_12a5 · 2022-10-25

**Confidence:** 4
**Correctness:** 2
**Technical Novelty And Significance:** 2
**Empirical Novelty And Significance:** 2
**Recommendation:** 3

**Clarity, Quality, Novelty And Reproducibility:**

The writing of the paper could be improved for a clearer presentation. The novelty is limited since the utilized techniques are very similar to other works.

**Strength And Weaknesses:**

Strength:
The authors proposed a one-stage training process training network from scratch without requiring a pre-trained network. The proposed method could achieve good performance.

Weaknesses:
1. There are many activation-aware pruning methods, whose idea is very similar to the proposed method. Thus, I am concerned about the lack of novelty in this paper.
2. The proposed method cannot compare with the latest SOTA methods. In fact, the proposed method cannot achieve the best performance compared with some recent works.
3. From the experimental results, one could see that the activation attention of the proposed method cannot gain much improvement.
4. The ablation study is actually a study of different activations. It is better to add the experimental results of the variant without activation.

**Summary Of The Paper:**

In this paper, the authors propose a filter-importance-scoring concept named pruning by active attention manipulation. The proposed method is a one-stage training process training network from scratch without requiring a pre-trained network.

**Summary Of The Review:**

There are many activation-aware pruning methods, whose idea is very similar to the proposed method. Thus, I am afraid that this work lacks novelty. The experiments are not solid to demonstrate the effectiveness and efficiency of the proposed method.

---

> ### Author Response · Authors · 2022-11-18
> **Clarifications**
>
> We thank the reviewer for their feedback on our work. In the following, we provide a point-by-point answer to the raised concerns. We hope that our response together with the discussions with other reviewers motivates a more positive score.
>
> **1)** We are not sure if we properly understood the reviewer’s comments about “activation-aware” pruning methods. In case the reviewer meant “training-aware” pruning methods, we would like to point out that PAAM is the first training-aware pruning method that learns the filter importance scores from the filter weights. Other methods do not do this and they assume that the scores are independent of each other.  For learning the scores from the filter weights, PAAM is also the first training-aware pruning method that uses attention for pruning. This allows PAAM to also extract the correlations and hidden patterns among filters.
>
> It would be great if the reviewer could point us to any related work on “activation-aware” so that we can provide more insights into how our method is the first of its kind.
>
> **2)** We would be glad to compare our results to the ones in other papers if the reviewer would provide us with the references of the “better SOTA” filter-pruning methods that they had in mind. We might have missed them. However, to the best of our knowledge, we have compared our results to all SOTA filter-pruning methods.
>
> **3)** We would thank the reviewer if they mention which tables they had in mind in their comment. As other reviewers have also mentioned, our results show that PAAM can achieve higher accuracy while pruning more parameters/flops.
>
> **4)** Attention layers use activation functions. Sigmoid is the most common in attention networks, as used for example in scaled Dot-Product Attention[1]. We introduce our custom activation function, leaky-expo, and compare its performance to Sigmoid in section A.2. Please note that the activation function is not the main contribution of PAAM. It is rather the addition of attention layers, that learn the scores from the filter weights. The activation function is only one component that helps to keep the scores nicely distributed in the desired range of 0-1.
>
> [1] Ashish Vaswani, Noam Shazeer, Niki Parmar, Jakob Uszkoreit, Llion Jones, Aidan N
> Gomez, Ł ukasz Kaiser, and Illia Polosukhin. Attention is all you need. Advances in Neural Information Processing Systems, 2017.

---

### Official Review · Reviewer_rXzi · 2022-10-31

**Confidence:** 4
**Correctness:** 3
**Technical Novelty And Significance:** 2
**Empirical Novelty And Significance:** 3
**Recommendation:** 8

**Clarity, Quality, Novelty And Reproducibility:**

Some claims in the paper may be somewhat misleading.

"PAAM can also train and generate a pruned network from scratch in a straightforward, one-stage training process without requiring a pre-trained network. "
This is somewhat misleading since you don't require a pre-trained network since your entire training process is rather more lengthy than a normal training process and ends with fine-tuning the pruned network to convergence.

**Strength And Weaknesses:**

Strengths:
- Achieves very strong possibly SOTA empirical results on CIFAR-10 and Imagenet.
- Proposes novel method for channel pruning using an attention mechanism that allows accounting of layer interdependencies.

Weaknesses:
- Little to no discussion of the stability or robustness of the approach.
- Possibly limited novelty due to the similarities to existing work in pruning and NAS and would benefit from significantly more analysis and ablation of their method and the cost.

Questions:
The paper would benefit from discussion of the cost of pruning compared to competing methods. Do other methods prune/fine-tune the network for as many epochs? How expensive is the score model?
Since the training method is quite lengthy and approaching NAS, it would be quite useful to run the pruning results multiple times to give a better idea of how stable the method is.
How much hyperparameter tuning did you conduct and do you have results from different weightings of the flops/parameter cost? The work would benefit from more discussion of how controllable the sparsity is.
Is a single score model trained used for every layer or are separate networks used per layer?
I don't understand the section on the threshold value. Table 3 shows that you get a more accurate, and sparser network with a value of 0.2 or 0.3. Why was 0.5 chosen?

Nits:
On page 7: "Table 2 compares the performance of PAAM (KQ) to the SOA on ImagNet" is missing Ts
Question, but a bit separate from the paper. Can the channel importance estimator transfer between models or during training or does it only give useful signal when co-trained with the network? Would be a useful area to explore if possible.


**Summary Of The Paper:**

The work proposes a novel method for network pruning by co-training a network to predict filter importance given model weights. This allows the model learn hidden correlations between the model weights and channel importance. They achieve strong empirical results on pruning resnet networks on CIFAR-10 and Imagenet. The method naturally learns E2E and doesn't require per-layer sparsity values and is tuned by a global cost-weighting/function. To satisfy the requirements of their attention network, they propose a novel activation function.

**Summary Of The Review:**

This work proposes a novel method for structured pruning which achieves significantly better SOTA results on Imagenet and Cifar10. Some aspects may of be limited novelty however, due to similarities to methods explored in other NAS and channel pruning methods. Since the empirical results are quite important, the work would benefit from more exploration of it's robustness and more detailed analysis of the method and training cost. I believe this work is currently marginally below the acceptance threshold, but would be willing to increase my score with additional  analysis, experimental results, and details of the hyperparameter search

---

> ### Author Response · Authors · 2022-11-18
> **Clarifications**
>
> We thank the reviewer very much for their detailed and constructive review of our work. In the following, we carefully address their raised concerns to encourage a more positive evaluation of our work:
>
> **The cost of pruning:**
> Finetuning is an important stage in all pruning methods. For instance, in Cifar10 experiments, CHIP uses a network pre-trained for 200 epochs, calculates the channel importance scores, and then fine-tunes the model for 300 epochs, which results in a total of **500** epochs. NPPM uses a network pre-trained for 200 epochs, trains the channel gates for 200 epochs, and then fine-tunes the model for 200 epochs, which makes it into a total of **600** epochs. Similarly, DMC uses a network pre-trained for 200 epochs, trains the channel gates for 300 epochs, and then fine-tunes the model for 160 epochs, which results in a total of **660** epochs. For the same experiments, PAAM starts with 50 warm-up epochs, followed by alternating training for 90 epochs, and then fine-tuning for 300 epochs, which makes it a total of **440** epochs.
>
> While fine-tuning plays an important role in improving performance, it is not the main cause of the high accuracy of PAAM compared to other methods. In order to better show this, we repeated our experiments with a smaller number of fine-tuning epochs. We start with 0 epochs, meaning no fine-tuning. In this case, we are training the model for only 50+90=140 epochs, which is even less than the 200 epochs needed to pre-train the base model. We then use 60 epochs to train the model for 200 epochs in total which brought PAAM to an accuracy of 93.44 which is already higher than the dense base model used in CHIP (with an accuracy of 93.26).
>
> Below is the table for the following numbers of fine-tuning epochs: 0, 60, 100, 200, and 300.
>
> | epochs   | 0     | 60    | 100   | 200   | 300   |
> |----------|-------|-------|-------|-------|-------|
> | Accuracy | 90.71 | 93.44 | 93.83 | 94.01 | 94.28 |
>
> **Hyper-parameter Tuning:**
> We did not perform any hyper-parameter tuning except for lambda as described below. We used 300 epochs for fine-tuning similar to CHIP. The number of epochs for the warm-up and alternating tuning was inspired by [1]. We used the same values for these hyper-parameters in all of our experiments. However, one could achieve even better results by searching for the optimum number of epochs for each phase.
>
> For the hyper-parameter lambda, we performed a grid search to gain the proper pruning ratios, since lambda is the only variable that controls the sparsity of the network. The higher the value of lambda is, the more parameters and flops are pruned.
>
> For the CNN learning rate, we did not perform hyper-parameter tuning and set it to 0.001. We used a lower learning rate for the AN because we were interested in a low convergence speed for the scores.
>
> **The different weighting of flops/parameters:**
> We were more interested in a balanced parameters/flop ratio achieved by simply adding the L1 norm of scores to the loss function. However, one could calculate the flops and parameters from the filter scores and use their values as a regularisation factor that is added to the loss function to weight parameters and flops differently by giving more weight to the desired factor.
>
> **Controllability of the sparsity:**
> The sparsity of the pruning is controlled by the value of lambda. One can use a lower/higher value for a lower/higher sparsity. Moreover, one can also use the threshold theta value for a more fine-grained sparsity control.
>
> **Single or separate networks per layer:**
> Each PAAM AN is associated with a CNN layer as the AN input consists of the filter weights of that CNN layer. Hence, each AN learns the specific correlations and patterns present in the filters of that CNN layer, which makes it unique. This also leads to different numbers of filters pruned in each CNN layer.
>
> **Threshold values table:**
> As Table 3 of the paper shows, we get slightly higher accuracies for threshold values lower than 0.5. However, a lower threshold than the one we chose also leads to lower parameters and flops pruned. If we chose the 0.3 threshold, we would get a 0.02 percent higher accuracy compared to 0.5 threshold, but we would also get 7 percent fewer flops and 2 percent fewer parameters pruned.
>
> **Transferability of the networks to other models:**
> Since PAAM learns the filter scores from the network weights, the scores learned are specific to that network only, meaning that transferring the ANs to other models would not be feasible. Moreover, the shape of the network weights (number of filters, kernel sizes) might be different in different layers/models, which makes it difficult to easily transfer between models.
>
> [1] Alexandra Peste, Eugenia Iofinova, Adrian Vladu, and Dan Alistarh. Ac/dc: Alternating compressed/decompressed training of deep neural networks. Advances in Neural Information Processing Systems, 34, 2021.

---

> > ### Author Response · Authors · 2022-11-18
> > **Stability and Robustness experiments**
> >
> > **Stability**
> > As suggested by the reviewer, we repeated one of our experiments for 5 times to show the stability of training. We repeated the experiment on Cifar10 with PAAM(KQ) with the average pruning ratio. Table below shows the mean and the variance of the results, alongside the values reported for the same experiment in the paper. As one can see, the results are pretty stable and with low variance.
> > |                  | Accuracy   | Params    | Flops     |
> > |------------------|------------|-----------|-----------|
> > |  Reported | 94.01      | 54.0      | 52.3      |
> > | 5 times repeated | 94.02±0.11 | 52.5 ±3.2 | 50.1 ±2.2 |
> >
> > **Robustness**
> > We would like to thank the reviewer for pointing out the importance of robustness evaluation since it is an important factor in network pruning. Research has shown that while pruned networks can maintain the accuracy of the dense network, they usually suffer when tested for robustness[1].
> >
> > Accordingly, we performed a new experiment as follows: In the table below, we show the robustness results of PAAM compared to CHIP where we corrupted the test images with Gaussian noise (with a zero mean) with increasing variances. One can understand these corrupted images as out-of-distribution test sets. Since there were no other robustness evaluations for SOTA, we restricted our comparisons to CHIP only (due to computing limits). In order to ensure fairness, we pruned the ResNet56 with CHIP with the same amount of parameters and flops pruned by PAAM.  We tested each model 5 times and report the mean and variance of the results.
> > | model | params | flops | clean | 0.01        | 0.05        | 0.1         | 0.15        |
> > |-------|--------|-------|-------|-------------|-------------|-------------|-------------|
> > | CHIP  | 52.3   | 49.3  | 93.46 | 93.37 ±0.04 | 92.37 ±0.13 | 88.44 ±0.12 | 81.37 ±0.24 |
> > | PAAM  | 52.3   | 49.3  | **94.28** | **94.08 ±0.02** | **93.00 ±0.10** | **89.35 ±0.13** | **82.18 ±0.21** |
> >
> > [1] Lucas Liebenwein, Cenk Baykal, Brandon Carter, David Gifford, and Daniela Rus. Lost in Pruning: The Effects of Pruning Neural Networks beyond Test Accuracy.  Proceedings of Machine Learning and Systems, 2021.

---

> > > ### Comment · Reviewer_rXzi · 2022-11-21
> > > **Increasing Score**
> > >
> > > I would like to thank the authors for their response to my questions. I am increasing my score and recommending acceptance of the paper since many of my concerns are addressed in the new experiments.
> > >
> > > I am still however a bit concerned how important the core change of this method of making the pruning model dependent on the results without a true ablation with everything else held the same. Since you are not sharing weights, it seems with enough training you should be able to get similar results without directly giving the model weights since there is no difference in model capacity given as long as the model weights are not changing, the model should still be able to overfit as there is no weight sharing. The paper would be much stronger with a direct ablation.

---

> > > > ### Author Response · Authors · 2022-11-22
> > > > **Thank you very much**
> > > >
> > > > We thank the reviewer very much for increasing their review score and voting for the acceptance of our paper. We are also very happy to hear that many of their concerns are addressed now.
> > > >
> > > > **Regarding a direct ablation analysis.** We would love to address this concern to motivate an even stronger acceptance of our paper. However, we have a hard time following what does the reviewer mean by a direct ablation study? Could you please explain what experiment would you like us to do so that we perform and make our potential camera-ready version stronger?
> > > >
> > > > Thank you, Authors

---

> > > > > ### Comment · Reviewer_rXzi · 2022-11-22
> > > > > **Effect of the weight conditioning**
> > > > >
> > > > > Hi Authors,
> > > > > To clarify, I mean the central contribution was that your model is dependent on the model weights. However, there were also a couple different choices made with the training pipeline and strategy. I meant that it would be useful to directly compare the model with access to the weights and without access to the weights, but trained in the same fashion. This could be done either giving the model a learned input that instead of the weights or just making the scores themselves trainable variables.

---

> > > > > > ### Author Response · Authors · 2022-11-22
> > > > > > **Thanks for Clarification**
> > > > > >
> > > > > > Thank you for clarifying this scheme you had in mind. This is a great suggestion, we will perform this ablation experiment and will provide results in the next 2 days.

---

> > > > > > > ### Author Response · Authors · 2022-11-24
> > > > > > > **Requested Ablation Study**
> > > > > > >
> > > > > > > As requested by the reviewer, we performed the following new ablation experiment.
> > > > > > >
> > > > > > > We pruned the ResNet56 network on Cifar10 with the PAAM pruning pipeline and loss function, but instead of learning the scores from the network weights, we set them directly as trainable parameters. We initialized all scores by 1, meaning we start with a dense network. Table below shows the results of this experiment compared to PAAM. As one can see, learning the scores from the network weights (PAAM) outperforms the version with trainable scores.
> > > > > > >
> > > > > > > | Model                               | Accuracy | params | flops |
> > > > > > > |-------------------------------------|----------|--------|-------|
> > > > > > > | PAAM vanilla                        | **94.28**    | **52.3**   | **49.3**  |
> > > > > > > | Trainable scores with PAAM pipeline | 93.74    | 50.0   | 48.4  |
> > > > > > >
> > > > > > > The network layer budgets discovered by the new model can be found [here](https://ibb.co/Mn0NSsS). As the plot shows, the version with trainable scores can still learn the bottleneck structure of layers similar to PAAM (Please see figure 4 in the paper).
> > > > > > >
> > > > > > > We plotted the histogram of the filter scores learned by both models [here](https://ibb.co/5MHNhYQ). (a) shows the histogram of the scores learned by PAAM, and (b) shows the same for the scores learned by the ablated model with trainable scores. Since in both methods most of the pruning happens in the even layers, we plotted the scores histogram for both methods, with odd and even layers separated [here](https://ibb.co/mJwgd6Z). As the histograms show, the model with trainable scores was not able to properly set the scores to values close to zero and one (Which is the desirable behavior). This is important as the scores are directly multiplied by the feature maps in the scores training phase (We have elaborated more on this in the manuscript Page 5 Section 2.5).
> > > > > > >
> > > > > > > The PAAM training pipeline and loss function help the model with trainable scores properly learn the intra-layer pruning budgets (which layers to prune more from). However, this model has a hard time in inter-layer pruning (which filters to prune in each layer). For the latter, learning the scores from the filter weights makes it significantly easier for the model to learn the scores. The filter weights have valuable information important for pruning, like the filter magnitude or the correlations among the filters.
> > > > > > >
> > > > > > > We once again thank the reviewer for suggesting this interesting ablation experiment to show that our entire PAAM compartments are indeed necessary for achieving better performance and satisfying the objective of the pruning scheme. We will include this experiment in our revised version. We hope that these results satisfy the reviewer's concerns and promote a clear acceptance of our paper.
> > > > > > >
> > > > > > > Sincerely,
> > > > > > >
> > > > > > > Authors

---

> > > > > > > > ### Comment · Reviewer_rXzi · 2022-11-24
> > > > > > > > **Activation function?**
> > > > > > > >
> > > > > > > > I appreciate the author's work to include these additional experiments and their details analysis.
> > > > > > > >
> > > > > > > > Did you use the activation function? If you did not, that would be the most obvious explanation why it has trouble with setting close to zero due to the reasoning for how you carefully designed it in the paper, if not, I would appreciate if you ran it that way since with these experiments with multiple changed it is easiest if you do a full ablation where you change only each singular aspect while keeping everything else the same. Especially when you male a couple carefully selected design choises like in this paper. Alternatively what I said in just making the model inputs trainable may be easiest way to show the distinction.
> > > > > > > >
> > > > > > > > If you run/already ran it that way and included those ablations in the full paper, I would be willing to raise my score.

---

> > > > > > > > > ### Author Response · Authors · 2022-11-25
> > > > > > > > > **Full Ablation**
> > > > > > > > >
> > > > > > > > > We performed the ablation with and without the leaky-expo activation function. The results above are for the version without the activation function. For the version with the activation function, we set all score parameters to zero to start with a dense network(exp(0)=1). This version resulted in a similar layer-budget discovery which can be found [here](https://ibb.co/NjbCP8R).  However, In this version, the scores mostly got stuck around 0.5 and did not converge to zero. [Here](https://ibb.co/1f0FMH3) you can see the histogram of the scores plotted, and [here](https://ibb.co/kgHJW3F) the histogram with even and odd layers separated for all three versions side-by-side. As the leaky-expo activation function was designed for the attention layers outputs, it is not well-suited for the version where the scores are directly trainable parameters themselves. In fact, in this case, it would be easier for the scores to reach values close to zero without the activation function. (The scores must be larger negative values so that their exponential gets close to zero).
> > > > > > > > >
> > > > > > > > > We will include these results in the full paper. We hope this addresses the reviewer’s concerns.
> > > > > > > > >
> > > > > > > > > Thanks,
> > > > > > > > >
> > > > > > > > > Authors

---

> > > > > > > > > > ### Comment · Reviewer_rXzi · 2022-11-27
> > > > > > > > > > **Strange results**
> > > > > > > > > >
> > > > > > > > > > It's understandable, but it seems like the architecture and choices may be more important than the access to the actual model weights which as you're pointing out to the other reviewers is the main technical novelties of this work compared to previous work. I appreciate the experiments you've done, but this seems to be a quite important part of your paper.
> > > > > > > > > >
> > > > > > > > > > Do you have time to just try substituting the model weights input with trainable parameters? That would directly show that the weights themselves are making a difference and not just gradient and parameterization issues causing the training to be difficult.

---

> > > > > > > > > > > ### Author Response · Authors · 2022-11-27
> > > > > > > > > > > **New Results**
> > > > > > > > > > >
> > > > > > > > > > > Of course! As instructed by the reviewer, we performed the following new experiment to fully address the raised issue:
> > > > > > > > > > >
> > > > > > > > > > > We defined the input trainable parameters for each layer with the shape of the layer weights, and fed these parameters to the attention layers instead of the model weights to train the filter scores. With this setting, the network was able to make the scores converge to values close to zero and one. You can find the histogram of the filters [here](https://ibb.co/7Q3WSvJ). Similar to the previous ablation models, this network also discovered the alternating layer budgets which can be found [here](https://ibb.co/9vWdnfX).
> > > > > > > > > > > However, in terms of the accuracy, the model could not catch up with PAAM. Table below shows the accuracy of both models before and after fine-tuning, alongside the pruning ratios.
> > > > > > > > > > >
> > > > > > > > > > > | Model                              | params | flops | Acc before fine-tuning | Acc after fine-tuning |
> > > > > > > > > > > |------------------------------------|--------|-------|-----------------------------|----------------------------|
> > > > > > > > > > > | PAAM vanilla                       | **52.3**   | **49.3**  | **90.71**                       | **94.28**                      |
> > > > > > > > > > > | PAAM vanilla with trainable inputs | 43.9   | 46.1  | 88.51                       | 93.79                      |
> > > > > > > > > > >
> > > > > > > > > > > As the table shows, the original PAAM achieves higher accuracy while pruning more parameters and flops. The difference in accuracies is especially apparent before the fine-tuning epochs and directly after pruning. These results show that while the convergence of the scores to zero and one is important for learning the proper scores, it is not the only reason why PAAM performs well. Having access to the model weights is essential for finding the best filter scores.

---

> > > > > > > > > > > > ### Comment · Reviewer_rXzi · 2022-11-27
> > > > > > > > > > > > **Question**
> > > > > > > > > > > >
> > > > > > > > > > > > Is Flops and Params the number of flops and parameters or the decrease in flops or parameters?

---

> > > > > > > > > > > > > ### Author Response · Authors · 2022-11-27
> > > > > > > > > > > > > **Answer**
> > > > > > > > > > > > >
> > > > > > > > > > > > > It is the percentage of the decrease in parameters and flops.

---

> > > > > > > > > > > > > > ### Comment · Reviewer_rXzi · 2022-11-27
> > > > > > > > > > > > > > **Increasing score**
> > > > > > > > > > > > > >
> > > > > > > > > > > > > > I thank the authors for the additional experiments and I will increase my score. I think the authors have demonstrated their method but with the caveat that I partially agree with the other reviewer. I think the paper would be significantly improved by the addition of these additional ablation experiments on Imagenet since results on CIFAR-10 are much less clear.

---

> > > > > > > > > > > > > > > ### Author Response · Authors · 2022-11-27
> > > > > > > > > > > > > > > **Thank you very much**
> > > > > > > > > > > > > > >
> > > > > > > > > > > > > > > We would like to thank the reviewer very much for greatly engaging with us during the discussion period. Their constructive comments have significantly improved our paper. We will make sure that these new ablation results be added to our final version.
> > > > > > > > > > > > > > > To fully address the other reviewer's concern as well, we are going to run the same ablation experiments instructed by Reviewer rXzi on ImageNet, and will certainly make sure to include those results in our final version, once available.

---

> > > > > > > > > > > > > > > ### Author Response · Authors · 2022-12-08
> > > > > > > > > > > > > > > **Results on ImageNet**
> > > > > > > > > > > > > > >
> > > > > > > > > > > > > > > Dear Reviewer,
> > > > > > > > > > > > > > >
> > > > > > > > > > > > > > > As requested by the other reviewer and promised, and to emphasize the effectiveness of PAAM, we have now obtained more results on ImageNet.  This is done to address the last concerns about ImageNet results and higher flops pruning ratio. To this end, we increased the flops pruning goal for ImageNet on ResNet50. The table below shows the results of PAAM compared to the references suggested by the reviewer.
> > > > > > > > > > > > > > >
> > > > > > > > > > > > > > > | Method  | accuracy | Pruned params | Pruned flops |
> > > > > > > > > > > > > > > |---------|----------|---------------|--------------|
> > > > > > > > > > > > > > > | IAP     | 74.54    | N/A           | 41.5         |
> > > > > > > > > > > > > > > | AAP (1) | 75.58    | N/A           | 31.88        |
> > > > > > > > > > > > > > > | AAP (2) | 75.07    | 37.17         | N/A          |
> > > > > > > > > > > > > > > | PGMPF   | 75.11    | N/A           | 53.5         |
> > > > > > > > > > > > > > > | PAAM    | **75.84**    | **49.3**          | **60.9**         |
> > > > > > > > > > > > > > >
> > > > > > > > > > > > > > > AAP and IAP refer to Adaptive activation-based structured pruning and Iterative activation-based structured pruning, respectively. PGMPF refers to Prior Gradient Mask Guided Pruning-Aware Fine-Tuning(PGMPF).
> > > > > > > > > > > > > > >
> > > > > > > > > > > > > > > As the results show, PAAM achieved considerably higher accuracy, while pruning more flops with a high margin compared to the baselines. We are now running the ablation experiment suggested by the reviewer on ImageNet too.
> > > > > > > > > > > > > > >
> > > > > > > > > > > > > > > We would like to kindly ask the reviewer to bring these results to the attention of the other reviewers and the AC, to advocate for the effectiveness of our results and a fair final decision on our paper.
> > > > > > > > > > > > > > >
> > > > > > > > > > > > > > > Thank you
> > > > > > > > > > > > > > >
> > > > > > > > > > > > > > > Authors

---

### Decision · Program_Chairs · 2023-01-20

**Decision:**

Reject

**Justification For Why Not Higher Score:**

Experimental advantages are not sufficiently demonstrated.

**Justification For Why Not Lower Score:**

N/A

**Metareview: Summary, Strengths And Weaknesses:**

This paper develops a method for pruning neural networks in which an auxiliary subnetwork learns to predict filter importance scores from filter weights.  After a discussion with substantial engagement between reviewers and authors, two reviewers are positive, while one reviewer maintains a reject rating.  The Area Chair shares some of the concerns raised by Reviewer 12a5 and views the central unresolved issue as follows:

The paper proposes a more sophisticated, yet more computationally expensive, method of learning which filters to prune.  However, almost all of the experimental results show minor differences in performance compared to competing (and possibly simpler) methods.  Review rXzi notes that "results on CIFAR-10 are much less clear" as Table 1 shows marginal benefits over previous publications.  In replying to Reviewer wYHx's comment that "proposed method was validated with only ResNets", the authors show results for VGG, but again there are at best minor improvements with respect to prior work.  Only ResNet50 results on ImageNet (in reply to Reviewer rXzi) hint at a nontrivial improvement.  But, one good result against a backdrop of more minor differences is short of a resounding experimental validation for a more complicated methodology.

With this context, the Area Chair is not comfortably brushing aside Reviewer 12a5's concerns and believes more work is required to build a broader (in terms diversity of tasks and network architectures) and stronger experimental case for the method.